# Mindfulness Training plus Nature Exposure for Veterans with Psychiatric and Substance Use Disorders: A Model Intervention

**DOI:** 10.3390/ijerph16234726

**Published:** 2019-11-27

**Authors:** William R. Marchand, William Klinger, Ken Block, Scott VerMerris, Tracy S. Herrmann, Crystal Johnson, Nicole Paradiso, Michael Scott, Brandon Yabko

**Affiliations:** 1VA Salt Lake City Health Care System, 500 Foothill, Salt Lake City, UT 84148 USA; William.klinger@va.gov (W.K.); tracysherrmann@gmail.com (T.S.H.); crystal.johnson8@va.gov (C.J.); michael.scott@va.gov (M.S.); brandon.yabko@va.gov (B.Y.); 2Park City Sailing Association, Park City, UT 84098, USA; ken.block@Parkcitysailing.org (K.B.); Scott.VerMerris@ParkCitySailing.org (S.V.); nicole@nicoleparadiso.com (N.P.); 3Department of Psychiatry, University of Utah, 501 Chipeta Way, Salt Lake City, UT 84108, USA

**Keywords:** veterans, mental disorders, mindfulness, substance-related disorders, complementary therapies

## Abstract

There is a need to develop novel complementary interventions aimed at enhancing treatment engagement and/or response for veterans with psychiatric and substance use disorders. There is evidence that both mindfulness training and nature exposure (MT/NE) may be beneficial for this population and that combining the two approaches into one intervention might result in synergistic benefit. However, to date, the MT/NE concept has not been tested. This article reports a pilot feasibility and acceptability study of MT/NE which was, in this case, provided via recreational sailing. The primary aim of this project was to develop a model intervention and evaluation process that could be used for future studies of MT/NE interventions using a variety of methods of nature exposure (e.g., hiking, skiing, mountain biking). Results indicate preliminary evidence that it is feasible to utilize MT/NE interventions for the population studied and that the MT/NE model described can serve as a template for future investigations. Further, there were significant pre- to post-intervention decreases in state anxiety, as well as increases in trait mindfulness. Three psychological instruments were identified that might be used in future studies to evaluate MT/NE outcomes. Results from this project provide a model MT/NE intervention template along with evaluation metrics for use in future studies.

## 1. Introduction 

The literature indicates high rates of psychiatric [1,2,3] and substance use disorders [4,5,6,7,8] among US military veterans. Even though effective conventional treatments exist, such as prolonged exposure therapy for post-traumatic stress disorder [9], not all veterans respond to treatment [10]. In fact, many veterans who complete treatment continue to have residual symptoms [9] and high treatment dropout has been reported [9]. For veterans with substance use disorders (SUDs), treatment challenges include resistance to enter and/or engage in treatment [11,12], partial effectiveness of current interventions [13], and high relapse rates [14]. Thus, there is a need to develop novel complementary interventions aimed at enhancing treatment engagement and/or treatment response among this population.

Mindfulness-based interventions (MBIs) have the potential to, at least partially, fill this gap. There is compelling evidence of the effectiveness of MBIs for psychiatric disorders [15,16,17,18,19,20,21,22,23] and SUDs [24,25,26,27,28,29,30,31,32] among the general population. There is also a developing literature describing benefits specifically for veterans [33,34,35,36,37,38,39,40,41]. However, in a recent study [42] our group found evidence of a high attrition rate among veterans for an eight-week evidence-based MBI (mindfulness-based cognitive therapy). Thus, treatment engagement may be a challenge for both conventional therapies and MBIs in this population.

Another line of investigation indicates evidence of positive outcomes from nature exposure adventure therapies in the general population [43,44] and among veterans [45,46,47,48,49], including a very limited literature [50,51,52,53] specifically regarding recreational sailing—which is the modality of nature exposure used in this investigation. While there are multiple possible methods (e.g., hiking, snowshoeing, etc.) that could be used for nature exposure, sailing was chosen in part because of our previous work [50] using recreational sailing as mechanism for nature exposure specifically with veterans. In that investigation, we found that recreational sailing was not associated with increases in anxiety or negative affect, but was associated with increased psychological flexibility. Additionally, participation was associated with a greater likelihood of successfully completing a residential substance abuse program. Finally, sailing is accessible for those with physical limitations, which many veterans experience.

The novel intervention described herein, mindfulness-based therapeutic sailing (MBTS), was developed in order to serve as model for combining mindfulness training with nature exposure (MT/NE). This concept was based, in part, upon the suggestion that in order to improve engagement, at least for SUD, treatments should go beyond a focus on eliminating substance use and expand to include experiences that will be enjoyable to clients [54]. Sailing in general has the potential to fill this role and we have previously provided evidence [50] that some veterans experience sailing as enjoyable. 

In addition to enhancing treatment engagement, it was posited that the experience of nature exposure itself might directly augment mindfulness training. As stated above, we have previously provided preliminary evidence [50] that nature exposure via sailing may be associated with increased psychological flexibility, which has a strong relationship with mindfulness [55], and there is some additional evidence that exposure to nature can enhance mindfulness [56]. Thus, it was hypothesized that MT/NE might provide a synergistic healing benefit from both the development of mindfulness skills and the benefits of nature exposure.

In summary, MBTS was developed as a model complementary MT/NE intervention with the ultimate aims of enhancing both treatment engagement and outcomes. This paper reports an initial pilot safety, feasibility, and acceptability study of MT/NE using MBTS as a model intervention.

To our knowledge, this is the first investigation of the concept of utilization of an intervention combining mindfulness training and nature exposure for military veterans with psychiatric and/or substance use disorders. Further, it is the first paper to describe a template and outcome metrics for future MT/NE investigations. Key findings indicate that for this population: (1) it is feasible and safe to utilize recreational sailing as a method of nature exposure; (2) it is feasible and acceptable to veterans to combine nature and mindfulness training; (3) instruments used in this investigation may be appropriate for future MT/NE studies, and; (4) preliminary results indicate that MBTS was associated with significant decreases in state anxiety and increases in trait mindfulness. These results suggest that further studies of MBTS for veterans with psychiatric and/or SUDs are warranted. More importantly, the model five-session MT/NE intervention and psychological instruments described herein can serve as a template for investigations that combine mindfulness training with any nature exposure activities (e.g., hiking, mountain biking, or skiing). Having such a template can facilitate consistency and comparability between various studies, as well as support comparisons of different methods of nature exposure such as mindfulness plus nature exposure versus mindfulness alone versus nature exposure alone. 

## 2. Materials and Methods 

MBTS was provided via a partnership between the Park City Sailing Association, a Park City Utah-based not-for-profit community sailing organization, and the Veterans Integrated Service Network 19 Whole Health Flagship site located at the Veterans Administration Salt Lake City Health Care System in Salt Lake City, Utah.

Five MT/NE sessions were provided, once per week over a five week period. The first and last sessions were classroom instruction only, while the other sessions included both classroom instruction and sailing. The first classroom-only experience included some basic sailing instruction and an introduction to mindfulness which lasted about one hour. The final classroom-only session included mindfulness instruction and an opportunity for participants to provide verbal feedback on the intervention. This session also lasted approximately one hour. 

Sessions two, three, and four occurred during the afternoon and lasted approximately three hours each. These took place on a reservoir near Park City, Utah. Each activity began with a half-hour introduction to sailing lecture and a safety briefing given by a US Sailing certified instructor. Participants had the opportunity to ask questions about sailing and safety procedures. After the introduction/briefing for sessions two and three, the veterans were split into two equal smaller groups which alternated between sailing and classroom mindfulness training for about one hour each. All veterans received the same intervention with both a sailing and mindfulness training component. For session four, all participants received mindfulness training together for about 30 min and then sailed together for about 1.5 h.

The mindfulness training was provided by a board-certified psychiatrist with an extensive personal mindfulness practice, who was also a trained and experienced mindfulness-based cognitive therapy (MBCT) teacher. The mindfulness training was developed by taking key elements of MBCT [57] and adapting those for this intervention. Participants were encouraged to practice mindfulness meditation during at least five of the seven days between classes for 10–15 min each day. The MBTS curriculum including the specific mindfulness training given in each session is outlined in Table 1.

Three boats were used for each sailing experience, skippered by US Sailing certified instructors. VA Salt Lake City Flagship site staff accompanied the veterans during the experience. All staff and participants wore personal flotation devices whenever they were on the boats and near the water. Veterans participated in the activity by taking turns at various positions on the boat under the guidance of the skipper. Additionally, participants assisted with keeping watch for other boats as well as leaving and returning to the dock. Finally, some participants assisted in the basic steps of rigging and derigging the sailboats, such as hoisting and dousing the mainsail and unfurling and furling the jib. The final sailing session included a race between participants’ boats. Each session concluded with a debriefing/discussion that lasted approximately one and a half hours. 

The participant recruitment process involved advertising with a flyer and by word of mouth. Participation was voluntary. No veterans were excluded from participating. Twenty-one veterans signed up to participate in the five-session intervention. 

One aim of this study was to evaluate preliminary outcomes as well as the potential psychological instruments for use in future studies of MT/NE. The State Trait Anxiety Inventory six-item short form (STAI: Y-6 item) [58,59], the Acceptance and Action Questionnaire II (AAQII) [60], and the Five Facet Mindfulness Questionnaire (FFMQ) [61] were administered pre- and post-intervention to evaluate changes in state anxiety, psychological flexibility, and trait mindfulness. Additionally, a locally developed post-intervention qualitative survey was utilized to evaluate the participants’ subjective experience of the intervention, and safety was informally evaluated by recording whether any injuries occurred. 

Data for this report includes pre- and post-intervention instruments and surveys completed by the participants, as well as additional information extracted from medical records. Some participants did not complete all instruments. Only subjects with a full data set were included in the analyses.

Data analysis consisted of paired, two-tailed t-tests to compare pre- and post-intervention instruments. Additionally, qualitative results of the locally developed survey are reported. 

This investigation was approved by the University of Utah Institutional Review Board (IRB)—which serves as the IRB for the VA Salt Lake City Health Care System—and by the facility Research and Development Committee.

This cross-sectional study utilized a retrospective collection of data initially gathered for clinical and program evaluation purposes. Data were also extracted from medical records, including demographic and diagnostic information. This investigation was approved by the IRB as a retrospective study, therefore the requirement for subjects to sign consent forms was waived. However, an indemnity form was signed for Park City Sailing Association.

## 3. Results

Participants were 21 veterans, 13 males and 8 females, with an age range of 31 to 70, (mean age = 55). Medical, psychiatric, and substance abuse diagnoses were obtained from a review of the electronic healthcare record. All of the participants had at least one psychiatric disorder. Sixteen (75%) had two or more psychiatric or SUD co-morbidities. The most common condition was PTSD but not all veterans (19, 91%) had this diagnosis. The next two most common disorders were any SUD (12, 63%) and major depressive disorder (9, 43%). Additionally, 19 veterans (91%) had one or more medical co-morbidities, with the most common being chronic pain (14, 74%) and hypertension (6, 29%).

Psychological changes were assessed by three instruments that were administered pre- and post-intervention (Table 2). Results indicated a non-significant increase in psychological flexibility as measured by the AAQ-II (mean increase from 41.82 to 45.09, *p* = 0.278). In contrast, there were significant pre- to post-intervention changes in state anxiety as measured by the STAI: Y-6 item (decrease in mean anxiety scores from 43.55 to 33.27 *p* =0.032), and mindfulness as measured by the FFMQ (increase in mean scores from 116.33 to 129.33, *p* = 0.047). 

## 4. Discussion

The primary aim of this pilot study was to investigate the feasibility and acceptance of a model MT/NE intervention among veterans with psychiatric and substance use disorders, and to do a preliminary evaluation of outcomes. This initial first step was necessary to develop a workable template for future more rigorous studies of MT/NE interventions for both veteran and wider community populations. A secondary aim was to evaluate the use of recreational sailing as a method of nature exposure for this population.

This pilot investigation provides evidence that MT/NE is a feasible intervention for the population studied. However, while the model intervention studied (MBTS) has the potential advantage of being accessible to those with physical limitations, the significant requirements for staff expertise and equipment create challenges that must be overcome. Thus, implementation of MBTS may only be feasible in certain situations, for example where a healthcare organization can partner with a sailing organization. Additionally, MBTS could be triggering for some veterans with PTSD, such as those with trauma exposure involving water. Future studies should screen for this possibility. As the field advances, it will be necessary to evaluate and compare various methods of nature exposure. The intervention described herein was developed specifically as protype that can be used for this purpose, as other nature exposure activities can be readily substituted for sailing.

Regarding acceptability, qualitative information from the post-intervention survey (Table 3) provides evidence that participants experienced combining nature exposure and mindfulness training to be both acceptable and pleasurable as 92% of respondents indicated that they enjoyed it, “A lot” or “Very much.” Interestingly, 100% endorsed that sailing was associated with feeling calm or relaxed, which is consistent with our previous results [50] from a study of nature exposure via sailing (which did not include mindfulness training). Seventy-five percent of respondents associated feeling calm or relaxed with the nature exposure. The most common suggestion (33%) for improving MBTS was to increase the time spent sailing. Even though most respondents enjoyed MBTS, the average number of sessions attended was three of the five. Given that one aim of the intervention was to enhance treatment engagement, it may be necessary to modify MBTS to enhance retention. Potential modifications include incorporating both sailing and mindfulness in all sessions, and increasing the amount of time spent sailing in each session.

In addition to evaluating feasibility and acceptability, this project also assessed preliminary outcomes as well as the potential of three psychological instruments for use in future studies of MT/NE interventions. Results indicated a non-significant increase in psychological flexibility as measured by the AAQ-II. There were significant pre- to post-intervention decreases in mean state anxiety as measured by the STAI: Y-6 item, and increases in trait mindfulness as measured by the FFMQ. Conclusions regarding cause and effect must be tentative given the small sample size and study design, however these results are encouraging. Further, these findings suggest these three instruments are appropriate for use in future investigations. Even though there was not a significant change in psychological flexibility, as was found in our previous study [50], there was a trend towards greater flexibility which indicates that this construct should be assessed in future more rigorous studies.

The informal evaluation of safety revealed that there were no accidents or injuries to veterans or staff during the five MBTS sessions, including three sailing activities. Further, there were no events that would have been likely to result in injury. Notably, 91% of participants had medical co-morbidities and, as in our previous study [50], some veterans had limitations of the ability to ambulate, but all were still able to participate fully in the sailing portion with minimal assistance from staff. Thus, MBTS may be a particularly useful intervention for individuals with physical limitations, such as difficulty ambulating, chronic pain, amputation, or loss of limb mobility that preclude or limit participation in some other nature exposure activities, such as hiking, skiing, or mountain biking. 

There are a number of limitations of this study. It was a cross-sectional study utilizing retrospective data collection, and the sample size was small and therefore the ability to detect statistically significant changes was limited. Participants were veterans who volunteered to participate and therefore there was selection bias. However, the aim of the project was to develop and test a model for use in future studies of MT/NE interventions for both veteran and wider community populations. This aim was achieved, and preliminary outcome data are encouraging.

As mentioned in the introduction, many veterans suffer from psychiatric and/or substance use disorders. Not all veterans respond to current treatments [10] and many responders continue to have residual symptoms [9]. Treatment engagement can be challenging for this population [9,11,12] and relapse rates are high [14]. Complementary MT/NE interventions such as MBTS have the potential to positively impact treatment engagement and outcomes for this population, and significantly reduce the burden of illness.

## 5. Conclusions

To our knowledge, this is the first paper to report an investigation of an MT/NE intervention for veterans with psychiatric and/or substance use disorders. Further, it is one of the very few investigations [50,51,52,53,62,63] into potential psychological or health benefits of recreational sailing. Results indicate that the specific MT/NE model tested (MBTS) is feasible and acceptable as an intervention for veterans with psychiatric illness and may be associated with decreased state anxiety and increased trait mindfulness. More rigorous studies of MBTS are warranted. More importantly, findings of this study provide preliminary proof of the concept that MT/NE interventions in general can be used for this population. Furthermore, a model for a five-session MT/NE intervention is provided along with potential instruments for evaluation that can be adapted to any nature exposure activity. This model provides a template that can be used by other investigators, such that results are comparable between studies and sites and which can be used for both veteran and community populations. Lastly, the model facilitates necessary comparisons of various methods of nature exposure, as the specific exposure method (e.g., rafting, hiking, sailing) can vary while the mindfulness components remain unchanged. At this point in time, it is unknown whether MT/NE interventions will ultimately prove to be either beneficial or cost-effective for veteran and/or other populations. However, there is enough theoretical and preliminary evidence for potential benefit that further studies are warranted. The intervention model and outcome metrics described herein provide a template that can inform and guide additional MT/NE studies.

## Figures and Tables

**Table 1 ijerph-16-04726-t001:** Mindfulness-based therapeutic sailing curriculum.

Session	Description	Duration	Topics
One	Classroom-only	1 h	Introductions Course overview and handouts Sailing briefing Introduction to mindfulness Mindfulness of breath exercise/inquiry Homework assigned
Two	Classroom and sailing	3 h	Mindfulness of breath exercise/inquiry Mindful awareness while sailing Review of homework Review of mindfulness versus autopilot Sitting meditation/inquiry Homework assigned
Three	Classroom and sailing	3 h	Mindfulness of breath exercise/inquiry Maintain mindful awareness while sailing Review of homework Review of mindfulness versus autopilot Sitting meditation/inquiry Introduce concept of pain × resistance = suffering Benefits of mindfulness Homework assigned
Four	Classroom and sailing	3 h	Mindfulness of breath exercise/inquiry Mindful awareness while sailing Review of homework Sitting meditation/inquiry Review concept of pain × resistance = suffering Review benefits of mindfulness Homework assigned
Five	Classroom-only	1 h	Mindfulness of breath exercise/inquiry Course review and feedback Review concept of pain × resistance = suffering Review benefits of mindfulness Review plans to continue mindfulness and sailing Closing

**Table 2 ijerph-16-04726-t002:** Participant responses to psychological instruments.

Instruments	Pre-Mean (SD)	Post-Mean (SD)	df	t	d
STAI: Y-6 item (n = 11)	43.55 (13.29)	33.27 (13.48)	10	2.49 *	0.75
AAQ-II (n = 11)	41.82 (8.45)	45.09 (4.81)	10	−1.15	−0.35
FFMQ (n = 12)	116.33 (14.24)	129.33 (17.57)	11	−2.24 *	−0.65

* *p* < 0.05.

**Table 3 ijerph-16-04726-t003:** Most common participant responses to key questions in the post-intervention survey.

Question	Percentage of Responses
What would make the class better?	More time sailing—33%
Was there anything about the sailing experiences that made you feel calm/relaxed?	Yes—100%
If yes, what exactly made you feel calm and relaxed?	Experiencing nature—75%
How much did you enjoy the entire class?	A lot or very much—92%

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
