# Peer review of "Mindfulness Training plus Nature Exposure for Veterans with Psychiatric and Substance Use Disorders: A Model Intervention"

_ijerph, 2019, doi:10.3390/ijerph16234726_

Round 1

Reviewer 1 Report

Please, try to edit the tables and follow APA 6.0 recommendations for presenting tables.

Author Response

Thank you for this suggestion.   All tables have reformatted and now follow APA 6.0 recommendations.

Reviewer 2 Report

This was a very interesting approach to interventions for veterans. The introduction is well written and this sets the tone for the rest of the paper. However, there are a few comments/questions I have: 

You mention that participants did not complete all instruments. How was the missing data dealt with in analysis? Did participants provide consent in writing? Was there an indemnity form for risk associated with participating in sailing? The socio-demographics of participants should form part of the results.  The first paragraph in the results section seems misplaced - it deals specifically with physical safety and is not a result. Would be better placed in the methods or discussion.  Table 2 please indicate significant values with an asterisk. And note what your p values are under your table.  Table 3 - please do not align center the text. I feel like this table could be presented in a better manner. Maybe even as a figure.  Is this not a cross sectional study rather than retropsective? 

Author Response

The authors thank the reviewer for the helpful comments.  Point-by-point responses to each comment are provided below.

---------------

You mention that participants did not complete all instruments. How was the missing data dealt with in analysis?

Only subjects with full data sets were included in the analyses.  This has been clarified in the “materials and methods” section.

Did participants provide consent in writing?

The subjects did not provide consent for the study in writing as this investigation was (as approved by the Intuitional Review Board) a retrospective study of data that was originally collected for program evaluation/clinical purposes.  The typical requirement for consent was waived by the IRB for this investigation. This has been clarified in the “materials and methods” section.

Was there an indemnity form for risk associated with participating in sailing?

Yes, this form was provided/required by Park City Sailing Association.  I have added this information to the “materials and methods” section.

The socio-demographics of participants should form part of the results. 

This information has been moved to the “results” section.

The first paragraph in the results section seems misplaced - it deals specifically with physical safety and is not a result. Would be better placed in the methods or discussion.

This information has been moved to the “discussion” section.

Table 2 please indicate significant values with an asterisk. And note what your p values are under your table.

We have made this change.

Table 3 - please do not align center the text. I feel like this table could be presented in a better manner. Maybe even as a figure.

We have modified this table in response to this comment and that of another reviewer.  If requested, the information could be changed to a figure presentation in a subsequent revision.

Is this not a cross sectional study rather than retrospective?

It is a cross sectional study but used data originally collected for clinical/program evaluation purposes but was then approved by the IRB to be used for research as a  retrospective study.  I have clarified this in the “materials and methods” and “discussion” sections.

Reviewer 3 Report

Overall really well done. I am a fan of the Oxford comma so I have that highlighted throughout. I also appreciate the study but I would like some more insight for how you controlled or assessed for if there was a trigger associated with the water or sailing. For instance in a Navy service member who may have anxiety depression or PTSD stemming from experiences on a Naval ship and may therefore not be inclined to participate, perhaps, or may have a more difficult time with the intervention. Your sample seemed to enjoy it but just a thought for going forward. Finally, the discussion is missing a lot for me in terms of why this matters and why you should continue on with future studies from this pilot

Abstract

line 22 should say "as a template"

Introduction

line 48: I still don't understand why sailing and more can be added in this paragraph to flush this out, even if the literature is limited. I would try to use this space to answer the question "why sailing" in response to literature on nature exposure. It could just be feasibility but you should state that to be clear line 57-59: Seeing this sentence earlier may help the question of why sailing. but it would still be helpful to better understand how this stacks up to a control group, even with preliminary data is it above and beyond mindfulness alone, nature exposure alone or more traditional exposure therapy? line 68: should be "Further, it is the first..." and the outcome metrics are unclear because this isn't a validation study you just want to determine if these measures are appropriate to see change? line 69: remove comma after that line 71: add comma after training

Methods

Overall, in this section is I was missing if there was there a consistent PTSD diagnosis? was their index event all combat related? How did you determine that water or sailing, for instance in Navy sailors, may not be another trigger

line 88: sessional should be session line 89: add comma after three line 90: I would lower case reservoir since it doesn't occur before the name of a specific reservoir  line 93-94: "the veterans were split.... mindfulness training" this part is confusing. Did they both get the same sailing and classroom mindfulness just in smaller groups or did one get sailing and the other mindfulness? line 118: the "(16, 75%)" stat is confusing where it's currently placed. You say all of the participants but you then say only 16/21 and if this is for the 2 or more than it belongs at the end of the sentence so as not to be confusing

Results

Good!

Discussion

line 156: Not sure I got the understanding of how safety was evaluated in this study just whether people got hurt. I'm not sure I would list it here as your primary aim considering you didn't have a formal evaluation of safety just non-injures line 156: add comma after feasibility line 163: "veterans had limitations" would be interesting to know what these limitations were. more assistance, less manual work, modification of equipment? line 166: add comma after skiing line 174: should be "as a prototype" line 178: "92 %" remove space between 2 and % line 184: add comma after feasibility line 187: for a discussion section I would cut out the numbers here and the research jargon since that can be found in the results line 191-192: I believe there's more to discern here from these results and small sample size shouldn't be your crutch considering this is a pilot so you want to be able to hang your hat on your effects since this is just the pilot. And I'm not sure that this just says these measures are appropriate when instead it shows that trait anxiety decreased which is a big deal and that mindfulness which is what we would expect since you're teaching them mindfulness. i would say more about why you may not have seen changes in psychological flexibility.  line 193: "retrospective" wouldn't it be longitudinal since you have pre and post data for each participant. Retrospective would be it was only recollection of ones experiences after an event.  line 196: "outcomes"  This is super unclear. see your first sentence in the conclusion. I thought you were examining the effectiveness of this intervention by evaluating the outcomes? You mentioned a lot about retention in the intro but didn't find enough time spent evaluating the retention rate of your intervention and in your discussion how that compared to other interventions in this population

Author Response

Overall really well done. I am a fan of the Oxford comma so I have that highlighted throughout. I also appreciate the study but I would like some more insight for how you controlled or assessed for if there was a trigger associated with the water or sailing. For instance in a Navy service member who may have anxiety depression or PTSD stemming from experiences on a Naval ship and may therefore not be inclined to participate, perhaps, or may have a more difficult time with the intervention. Your sample seemed to enjoy it but just a thought for going forward. Finally, the discussion is missing a lot for me in terms of why this matters and why you should continue on with future studies from this pilot

We have addressed these recommendations to the extent possible including adding information about “why this matters” to the “discussion” section.

Abstract

line 22 should say "as a template"

We have made this correction.

Introduction

line 48: I still don't understand why sailing and more can be added in this paragraph to flush this out, even if the literature is limited. I would try to use this space to answer the question "why sailing" in response to literature on nature exposure. It could just be feasibility but you should state that to be clear line 57-59: Seeing this sentence earlier may help the question of why sailing. but it would still be helpful to better understand how this stacks up to a control group, even with preliminary data is it above and beyond mindfulness alone, nature exposure alone or more traditional exposure therapy? line 68: should be "Further, it is the first..." and the outcome metrics are unclear because this isn't a validation study you just want to determine if these measures are appropriate to see change? line 69: remove comma after that line 71: add comma after training

We have expanded the introduction to provide more information regarding, “why sailing.” The current literature does not address the question of nature exposure alone, versus mindfulness alone versus the two combined.  However, at the end of the “introduction,” we indicate that our work could lay the groundwork for such investigations.  We have added the actual changes in anxiety and mindfulness to the outcomes to indicate we were interested in both the appropriateness of the instruments and the actual results.

We have also corrected errors mentioned.

Methods

Overall, in this section is I was missing if there was there a consistent PTSD diagnosis? was their index event all combat related? How did you determine that water or sailing, for instance in Navy sailors, may not be another trigger?

In response to another reviewer, this information was moved to the “results” section.  We have clarified that most, but not all, participants had PTSD.  We did not screen for the possibility that water or sailing could be a trigger.  We have added the point that this should be done in future studies to the “discussion” section.

line 88: sessional should be session line 89: add comma after three line 90: I would lower case reservoir since it doesn't occur before the name of a specific reservoir  line 93-94: "the veterans were split.... mindfulness training" this part is confusing. Did they both get the same sailing and classroom mindfulness just in smaller groups or did one get sailing and the other mindfulness? line 118: the "(16, 75%)" stat is confusing where it's currently placed. You say all of the participants but you then say only 16/21 and if this is for the 2 or more than it belongs at the end of the sentence so as not to be confusing

We have clarified that all Veterans received the same intervention, with both sailing and mindfulness training. We have made the other recommended changes.

Discussion

line 156: Not sure I got the understanding of how safety was evaluated in this study just whether people got hurt. I'm not sure I would list it here as your primary aim considering you didn't have a formal evaluation of safety just non-injures line 156: add comma after feasibility line 163: "veterans had limitations" would be interesting to know what these limitations were. more assistance, less manual work, modification of equipment? line 166: add comma after skiing line 174: should be "as a prototype" line 178: "92 %" remove space between 2 and % line 184: add comma after feasibility line 187: for a discussion section I would cut out the numbers here and the research jargon since that can be found in the results line 191-192: I believe there's more to discern here from these results and small sample size shouldn't be your crutch considering this is a pilot so you want to be able to hang your hat on your effects since this is just the pilot. And I'm not sure that this just says these measures are appropriate when instead it shows that trait anxiety decreased which is a big deal and that mindfulness which is what we would expect since you're teaching them mindfulness. i would say more about why you may not have seen changes in psychological flexibility.  line 193: "retrospective" wouldn't it be longitudinal since you have pre and post data for each participant. Retrospective would be it was only recollection of ones experiences after an event.  line 196: "outcomes"  This is super unclear. see your first sentence in the conclusion. I thought you were examining the effectiveness of this intervention by evaluating the outcomes? You mentioned a lot about retention in the intro but didn't find enough time spent evaluating the retention rate of your intervention and in your discussion how that compared to other interventions in this population

We have clarified in the “materials and methods” and “discussion” sections that safety was evaluated informally by recording whether any injuries occurred.  We have also removed safety evaluation as a primary aim. 

We have made minor modifications to the abstract as well as the “introduction” and “discussion” sections of the main manuscript to better highlight the pre- to post changes demonstrated by the psychological instruments.

The retention rate in this study and implications for MBTS as an intervention have been added to the discussion section.

Another reviewer also commented on the use of “retrospective” and suggested the term, “cross-sectional.”  We have modified the manuscript to use that terminology while clarifying that data collection occurred pre- to post- for clinical/program evaluation purposes, but the IRB approval was to use that same data at a later date for research purposes by way of a retrospective study approval.  If necessary, this could be further clarified and/or the use of “longitudinal” substituted in a subsequent revision.

We have made the other recommended changes.